# Listens like mel: boosting latent audio diffusion with channel locality

## Abstract

Latent representations play a critical role in diffusion-based audio generation. We observe that Mel-spectrograms exhibit an approximate power-law spectrum that naturally aligns with diffusion's coarse-to-fine denoising process, whereas waveform variational autoencoder (VAE) latents display nearly uniform energy across channels. To bridge this gap, we introduce *channel-span masking*, an operation that, in expectation, behaves like a rectangular window over channels and thus acts as a low-pass filter in the channel–frequency domain, increasing channel locality. The induced locality steepens the latent spectral slope toward a power-law distribution and yields up to 2–4× faster convergence of Diffusion Transformer (DiT) training on audio generation tasks, while preserving reconstruction fidelity and compression ratio. Experimental results show that our model performs on par with, or better than, competitive baselines under the same conditions. Our codes are available at https://anonymous.4open.science/r/lafa-F2A2

## 1 Introduction

The Mel-spectrograms has long been the representation of choice for audio generation tasks, especially when incorporated with diffusion models (Le et al., 2023; Forsgren & Martiros, 2022; Liu et al., 2022; Zhu et al., 2023). This preference seems unusual in deep learning, where hand-crafted features are typically overshadowed by learnable counterparts, yet Mel-spectrograms offer two key advantages. First, it enjoys promising interpretability with semantic richness; Second, it is capable to represent frequency components in decibel scale, aligning with human perceptual biases and the inductive priors of generative models.

Diffusion models usually follow a coarse-to-fine synthesis paradigm(Rissanen et al., 2022; Falck et al., 2025). Natural images exhibit approximate power-law power spectral density (PSD) of the form $1/f^\alpha$, and a similar regularity has also been observed in Mel-spectrograms(Haro et al., 2012b). Under Gaussian noising, the forward process preferentially corrupts high frequencies below long frequency components, enabling the reverse process to reconstruct low-frequency structures first and progressively add fine details. This spectral ordering complements Mel's inherent low-frequency emphasis, directing diffusion capacity to regions most sensitive to human perception.

However, in long-form audio generation scenarios, high-resolution Mel-spectrograms are often required for high-fidelity reconstruction, *e.g.* 44.1 kHz, which poses challenges for existing generative models. Latent diffusion models mitigate this by leveraging VAEs to map signals to compact latent spaces, enabling efficient denoising(Rombach et al., 2022). Yet, as shown in Figure 2, we observe that compressed audio latents often exhibit amplified high-frequency energy along the channel axis, possibly deviating from the Mel power-law bias and also undermining the spectral autoregression

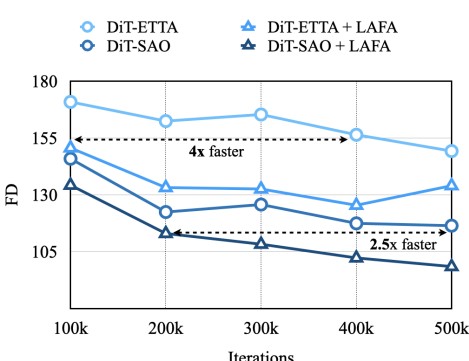

Figure 1: Comparison of convergence speed between a vanilla VAE and a VAE with LAFA, using SAO-DiT and ETTA-DiT on the SongDescriber dataset.

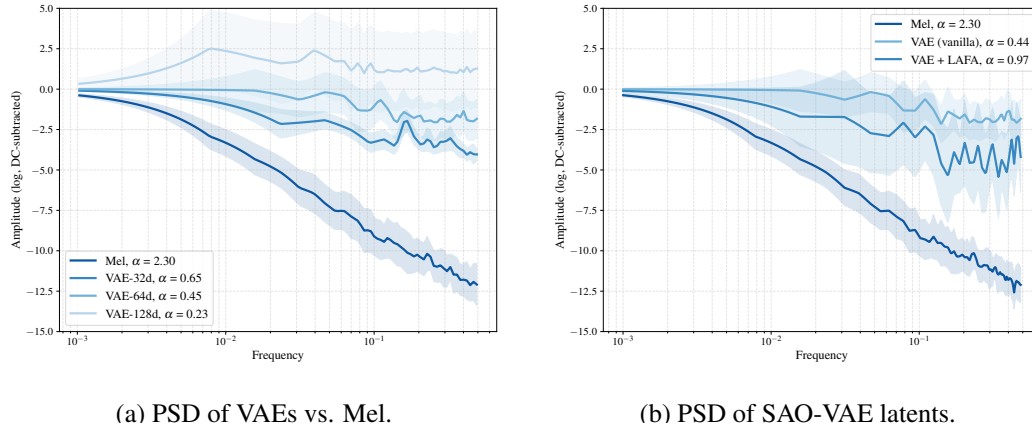

(a) PSD of VAEs vs. Mel.  (b) PSD of SAO-VAE latents.

Figure 2: (a) Power spectral density (PSD) of Mel spectrogram vs. VAE latent. Mel exhibits a power-law decay along the channel axis, whereas VAE latent retain more high-frequency energy. (b) PSD of SAO-VAE latent vs. LAFA latent: LAFA suppresses high-frequency components, yielding a PSD closer to a Mel spectrogram. The slope $\alpha$ is reported as the spectral decay exponent.

essential for diffusion models. This motivates us to pursue the elaborate design of a representation that unifies mel-like structural inductive bias with VAE compression efficiency.

We conduct a visualization-based analysis of VAE latents and find that certain channels encode pure noise, which not only degrades reconstruction fidelity but also introduces high-entropy information that hinders diffusion modeling. This phenomenon becomes more pronounced as the channel dimensionality increases, suggesting that larger-capacity VAEs do not necessarily yield better generative performance.

Mask modeling has been widely used as a self-supervised learning strategy to enhance latent encoders. Typically, masking is applied along the time axis for audio, or on patches for images and videos. The task requires the decoder to reconstruct masked tokens from neighboring unmasked tokens, thereby encouraging the encoder to produce smoother latents that facilitate prediction.

Motivated by this insight, we introduce **La**tent **F**low M**A**E (LAFA), a masked VAE bottleneck. Specifically, we apply span masking along the latent channel dimension, with a causal mask to impose monotonic order along the channel axis. Our experiments show that this approach improves local correlations across channels, revealing structured locality in the latent representation. Theoretically, we demonstrate that mask modeling functions as a low-pass filter in the spectral domain. We show that channel-span masking acts as a rectangular window in the channel axis, equivalent to a convolution in the frequency domain.

**Contributions.** (i) Analysis. We provide the first systematic study of the channel axis in audio VAEs, framing it as an object of geometric and spectral design for diffusion. To our knowledge, this is the first theoretical treatment of mask modeling as a low-pass filter. (ii) Method. We propose LAFA, a simple and plug-in VAE bottleneck that performs masked latent modeling with a flow-based latent decoder, producing clean latents for downstream training. LAFA combines the spectral bias of Mel-spectrograms with the compression capacity of a VAE, serving as an advanced alternative to Mel features. (iii) Empirics. We present comprehensive evaluations on audio and music generation tasks. LAFA improves sampling quality, accelerates DiT convergence by 2–4×, and generalizes effectively across DiT architectures and diffusion objectives.

## 2 BACKGROUND

### 2.1 TEXT TO AUDIO SYSTEM

*Non-autoregressive (NAR)* systems generate in parallel and model signals more globally via iterative refinement in diffusion models. Spectrogram-space models (e.g., Riffusion (Forsgren & Martiros,

2022), MusicLDM (Chen et al., 2023)) denoise directly in the time–frequency domain, while latent-diffusion systems first compress audio into a continuous latent with a variational autoencoder (VAE) and perform denoising there before decoding to waveform, as in AudioLDM(Liu et al., 2023),AudioLDM2 (Liu et al., 2024), Stable Audio Open (Evans et al., 2025), Noise2Music (Huang et al., 2023), Moûsai (Schneider et al., 2023), and two-stage pipelines like MusicFlow that bridge semantic units to acoustic latents (Prajwal et al., 2024). In the latent-diffusion regime, the VAE is more than a compressor; it is a design lever. Its channel count and frame rate, the choice of prior distribution and the strength of regularization, and the latent-space geometry they induce (e.g., locality and spectral bias) materially affect the trainability of the denoising backbone, the speed of convergence, and the final fidelity achievable under a fixed compute budget. Consequently, contemporary systems often operate directly in a VAE latent space without extensive discussion of representation design.

## 2.2 REGULARIZATION OF VAE

VAE regularization significantly shapes the latent space. Starting with LDM(Rombach et al., 2021), the KL loss has been commonly used to constrain the latent space, assisting in the supervision of the perceptual model and maximizing channel decoupling, closer to perception. VA-VAE(Yao et al., 2025) explored various perceptual models in detail. Furthermore, it was discovered that the KL loss does not necessarily preserve the information most conducive to generation. EQ-VAE(Kouzelis et al., 2025) and SE(Skorokhodov et al., 2025) proposed constraining the high-frequency components of the representation space by aligning to low-frequency information. DITO(Chen et al., 2025) removes KL divergence and uses noise perturbation as a bottleneck.

## 2.3 MASKED SELF SUPERVISED LEARNING

Channel regularization in audio VAEs is relatively underexplored, so we draw on insights from masked self-supervised learning (SSL). Along the *time* axis, wav2vec (Schneider et al., 2019) and wav2vec 2.0 (Baevski et al., 2020) predict masked/future spans in a *unidirectional* setup; subsequent works emphasize *bidirectional* context, e.g., HuBERT and BEST-RQ (Chiu et al., 2022) use temporal span masking to predict hidden units from both sides of the context. In the *time–frequency* regime, AudioMAE (Huang et al., 2022) and BEATs (Chen et al., 2022) extend masked autoencoding to spectrograms, learning joint time–frequency structure with high-ratio masking, inspired by the vision MAE framework (He et al., 2021).

Although masked SSL was developed for understanding tasks, its representations are now widely used in generation pipelines: w2v-BERT (Chung et al., 2021) style semantic units are adopted as intermediate targets/tokens in AudioLM-like systems (Liu et al., 2024); HuBERTHsu et al. (2021) representations supervise the first stage in MusicFlow (Prajwal et al., 2024); and features derived from AudioMAE are used as semantic guidance (Liu et al., 2024). Because SSL focuses on semantic representation rather than exact reconstruction, many systems employ a *two-stage* design: learn semantic tokens with SSL, then map semantics to acoustic latents. Recent work further distills SSL features into tokenizers/codecs to stabilize generative training (Ye et al., 2025; Ahasan et al., 2024).

## 3 SPECTRAL BIAS AND LOCALITY OF LATENT SPACE

How do spectral distributions differ between real-world audio representations and VAE latents, and how do these differences affect diffusion model convergence? In Section 3.1 we analyze the spectral statistics of Mel features and VAE latents, showing that Mel features exhibit a power-law spectral bias, whereas VAE latents deviate from this trend with disproportionately large high-frequency components. In Section 3.2 we relate these phenomena to latent locality, showing via correlation structure that Mel features are more locally correlated than VAE latents, especially as latent dimensionality grows.

### 3.1 POWER-LAW SPECTRAL BIAS

The data distribution shapes both the forward noising process and the learned reverse dynamics of diffusion models (Ho et al., 2020) and is fruitfully examined through its spectral statistics (Wang & Pehlevan, 2025). Power-law structure is ubiquitous across natural signals-images (Torralba & Oliva,

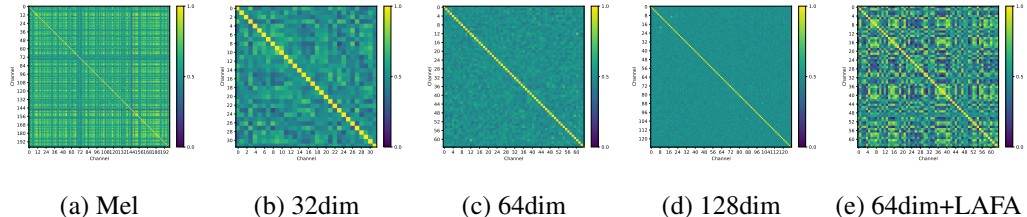

|  (a) Mel | (b) 32dim | (c) 64dim | (d) 128dim | (e) 64dim+LAFA |

Figure 3: Channel correlation heat maps of different features. (a–e) show pairwise channel correlations for Mel-spectrograms and VAEs. As the VAE channel dimension increases, channel locality decreases and deviates from the Mel-spectrogram structure. LAFA, illustrated on the 64-dim VAE, restores locality in the latent space. Locality is quantified by the Local Predictability Index (LPI): (a) Mel-spectrogram, LPI = 0.42; (b) 32-dim VAE, LPI = 0.26; (c) 64-dim VAE, LPI = 0.10; (d) 128-dim VAE, LPI = 0.09; (e) 64-dim VAE + LAFA, LPI = 0.30.

2003), audio(Torralba & Oliva, 2003), and video (Attias & Schreiner, 1997) and prior work reports a power-law spectral bias in Mel (Haro et al., 2012a; Dieleman, 2024). In contrast, the spectral properties of compressed waveform latents remain underexplored.

We estimate the spectral decay exponent by fitting a power law to the latent power spectrum. Concretely, given the radially averaged spectrum $S(f)$, we discard the DC bin, move to log–log coordinates, and perform a least-squares linear fit $\log S(f) \approx -\alpha \log f + b$; the slope $\alpha$ is reported as the spectral decay exponent. A complete description of the power spectrum computation is provided in the Appendix C.

For Mel features, $X$ is the log-magnitude spectrogram (time $\times$ Mel), and we compute the 2-D DFT and radial averaging per clip to obtain $S(f)$, where low $f$ reflects slow variations over time and Mel and high $f$ captures fine fluctuations; for VAE latents, $X$ is the (time $\times$ channel) latent map, and we apply the same 2-D DFT and radial averaging independently for each batch item to obtain $S(f)$ in the same way.

We analyze the frequency profiles of SAO-type VAEs trained for 1M steps, which is a regime where latent standard deviation and validation metrics stabilize. For each model we average spectra over 200 clips and multiple temporal windows. Figure 2 compares Mel space and VAE latents: (i) Mel features follow a power law with $\alpha \approx 2$ for both amplitude and aggregate variance across windows; (ii) VAE latents exhibit comparatively inflated high-frequency mass, increasingly deviating from the power-law slope as the channel count grows.

With an isotropic $\mathcal{N}(0, I)$ prior, increasing the channel count gives the encoder more axes along which to distribute information while remaining close to a factorized prior. The optimization therefore favors decorrelated, near-independent channels, yielding rapidly varying features across channels and elevating graph-high-frequency components. This complicates the latent geometry and, we hypothesize, undermines the desired coarse-to-fine learning dynamics in diffusion.

### 3.2 LOCALITY OF LATENT SPACE

We posit that channel locality—nearby channels carrying related information—helps preserve the power-law spectral bias characteristic of natural signals.

**Local Predictability Index (LPI$_\lambda$)** For each channel $c$, predict it from its $\pm r$ neighbors(where $r = \lceil \lambda C \rceil$) using ridge regression over all samples. Higher LPI$_\lambda$ indicates more local redundancy.

**Definition.** With latents $z \in \mathbb{R}^{B \times C \times T}$, flatten samples $N = B \cdot T$. For each $c$,

$$\text{LPI}_\lambda(z) \;=\; \frac{1}{C} \sum_{c=0}^{C-1} R^2\Big( z_c \;\leftarrow\; \{z_{c-r}, \ldots, z_{c-1}, z_{c+1}, \ldots, z_{c+r}\}\Big), \qquad r = \lceil \lambda C \rceil.$$

Using 200 samples, with $\lambda$ as 0.06, we compute channel-wise correlations and plot channel-correlation heat maps. Figure 3 shows: i) Mel features exhibit strong local correlation along the

Mel-bin axis. ii) A 32-dimensional VAE attains the highest LPI, with a block-local structure most reminiscent of Mel. iii) As channel count increases, VAE correlations diminish and LPI drops.

Due to compression pressure from the reconstruction loss, lower-dimensional latents enforce heavier feature reuse across nearby channels, promoting locality. Higher-dimensional latents reduce redundancy by decorrelating channels to represent distinct factors, which improves pure compression but degrades locality and disrupts the latent manifold's smooth structure-undesirable for continuous semantics and for stable, hierarchical generation in diffusion models.

## 4 METHOD

Motivated by the observations in Section 3, we introduce LAFA, a plug-in bottleneck that reshapes an existing VAE's latent bottleneck using channel-mask modeling. In masked autoencoders (MAE), a context encoder processes masked inputs and a decoder predicts the masked latents. Unlike classical MAEs that mask the origin signal, LAFA masks latent space, directly learning pressure toward abstract latent structure while avoiding costly training on brittle waveform details. Figure 4 outlines the approach.

### 4.1 CHANNEL-MASK MODELING

Along the time axis, audio exhibits strong short-range temporal dependencies. Along the channel axis, however, pronounced locality typically arises only under high compression (see Section 3.2). In higher-dimensional VAEs, channels tend to decorrelate, yielding a brittle, highly oscillatory latent geometry that complicates generative prediction. To increase locality and regularity across channels, we impose a mask-modeling task on the channel axis.

Mask ratio and spatial distribution are critical for self-supervised efficiency. If the mask ratio is too small, the task is trivial and the encoder learns little (Bardes et al., 2024; Song et al., 2025). If masks are uniformly scattered point-wise across channels, a decoder can rely on near-neighbor interpolation, again weakening pressure to learn abstract structure. Inspired by BERT-style span masking, we replace uniform per-channel masking with contiguous masked spans, forcing the decoder to rely on distant channels and thereby strengthening channel-wise locality in the learned representation.

Specifically, we let $C$ be the channel dimension and $\rho \in (0,1)$ the target mask ratio. The total masked length is $L_{\text{mask}} = \lfloor \rho\, C \rfloor$. Then, sample a span count $M \sim \text{Uniform}\{1,\dots,m\}$. $m$ is the hyperparameter of the max span count. Partition $L_{\text{mask}}$ into $M$ spans with random lengths that sum to $L_{\text{mask}}$, each truncated to at most $\lceil L_{\text{mask}}/M \rceil$. Start indices are randomly placed while there is no overlap between spans. This stochasticity prevents the encoder from overfitting to a fixed masking pattern.

Following Evans et al. (2025), we optionally impose a causal attention mask along the channel index, encouraging reliance on global context while enforcing a monotone ordering over channels. In addition, the model masks all channels with a certain probability. This training strategy allows the model to learn unconditional generation and achieve better generalization.

### 4.2 THEORETICAL ANALYSIS OF SPAN MASKING

**Proposition 1** Let $S$ be a circulant low-pass with DFT $H(k)$. For masks $M$ that zero a uniformly random length $w$ contiguous span and inpainting $A_M z = M z + (I - M)S z$, the expected pretext loss equals

$$\mathbb{E}_M \|A_M z - z\|^2 = \sum_{k=0}^{C-1} \lambda_k |z_k|^2, \qquad \lambda_k \approx \frac{w}{C}|1 - H(k)|^2,$$

i.e., a frequency-weighted quadratic form that penalizes high channel frequencies.

**Proof of Proposition 1** Let channels be a cyclic 1-D grid of length $C$; at each time $t$ we work with $z \in \mathbb{R}^C$. A mask $M$ is diagonal with 1 on observed (unmasked) entries and 0 on a contiguous span of length $w$. A (linearized) inpainting map is $A_M$ so that

$$\hat{z} \approx A_M z, \qquad \mathcal{L} = \mathbb{E}_M \|A_M z - z\|_2^2 = z^\top \Gamma z, \quad \Gamma := \mathbb{E}_M\big[(A_M - I)^\top (A_M - I)\big] \succeq 0.$$

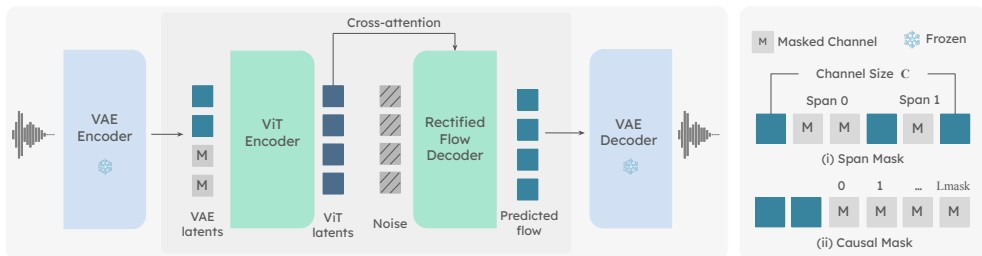

Figure 4: LAFA Architecture and Masking Strategy. **Left**: training pipeline. A frozen VAE encoder maps audio to latents; contiguous *channel spans* M are (optionally causally) masked and fed to a ViT encoder, whose output conditions a rectified-flow decoder via cross-attention to inpaint the masked channels. A frozen VAE decoder then reconstructs the audio. **Right**: illustration of span masking and causal masking along the channel axis.

Thus, training the encoder minimizes $\mathbb{E}_x[z(x)^\top \Gamma z(x)]$, i.e., it suppresses the eigendirections of $\Gamma$ with large eigenvalues. If $\Gamma$ is diagonal in the channel-DFT basis, then the eigenvectors are the channel Fourier modes and the eigenvalues are frequency weights.

Let $S$ be a fixed circulant smoothing operator (e.g., moving average of width $r$ or a triangular smoother). Define the inpainting:

$$A_M z = Mz + (I - M)Sz \quad \Rightarrow \quad A_M - I = (I - M)(S - I).$$

Then

$$\Gamma = \mathbb{E}_M\big[(I - M)(S - I)^\top (S - I)(I - M)\big].$$

For span locations drawn uniformly over the ring, $\mathbb{E}_M[I - M] = pI$ with $p = w/C$, and by rotational symmetry the expectation above is circulant. A standard calculation (or replacing $(I - M)$ by its mean $pI$ which becomes exact as spans are uniformly distributed over many steps) yields the tight approximation

$$\Gamma \approx p(S - I)^\top (S - I).$$

Since $S$ is circulant, it has DFT response $H(k)$. Therefore the DFT basis vectors $u_k$ are eigenvectors of $\Gamma$ with eigenvalues

$$\lambda_k \approx p|1 - H(k)|^2.$$

Key property. For any low-pass $S$, $|H(k)|$ is close to 1 at small $k$ and decays with $k$. Hence $|1 - H(k)|$ is small near $k = 0$ and large at high $k$. So high channel frequencies are penalized more, pushing the clean latent distribution toward channel-smooth signals. Hence longer spans ($w$ larger) amplify the preference for smooth (low-$k$) latents.

### 4.3 MODEL ARCHITECTURE

LAFA augments a pretrained VAE with: i) a ViT-style latent encoder operating on masked channels and ii) a Rectified Flow latent decoder.

**Encoder** A frozen VAE encoder maps a stereo waveform $x \in \mathbb{R}^{T \times 2}$ to latents $Z \in \mathbb{R}^{N \times C}$, where $N$ is the latent frame count and $C$ the channel dimension. Latents are mapped to hidden dim through Linear layer, and then position embedding is added to latents. The ViT encoder consumes the masked latents $\tilde{Z}$ and outputs contextual representations $h \in \mathbb{R}^{N \times H}$.

**Decoder** Referring to the original MAE, using the VIT decoder and trained with MSE, the reconstructed audio is blurry, with blurred harmonics. We attribute this to the blurring nature of the MSE loss and the high mask ratio. In contrast, LAFA uses flow matching as the reconstruction objective and can achieve reconstruction results close to the original VAE at high mask ratios. in latent space, conditioned on $h$ via cross-attention. Let $Z_0$ denote the clean latents and $\varepsilon \sim \mathcal{N}(0, I)$. We sample $t \sim \mathcal{U}[0, 1]$ and form $Z_t = (1 - t)Z_0 + t\varepsilon$. The decoder predicts the vector fields $U = \varepsilon - Z_0$ conditioned on $(Z_t, h, t)$.

Table 1: Generation performance on SongDescriber and AudioCaps. **Stage I** toggles LAFA fine-tuning of the SAO-VAE latents (*SAO-VAE* = vanilla; + *LAFA* = with LAFA). **Stage II** trains a Diffusion Transformer with either v-pred ($v$-parameterization) or RF (rectified flow).

| Stage II Model / Objective | Stage I | SongDescriber (Music) | | | AudioCaps (Sound) | | |
|---|---|---|---|---|---|---|---|
| | | $\text{CLAP}_{\text{score}} \uparrow$ | $\text{KL}_{\text{PaSST}} \downarrow$ | $\text{FD}_{\text{openl3}} \downarrow$ | $\text{CLAP}_{\text{score}} \uparrow$ | $\text{KL}_{\text{PaSST}} \downarrow$ | $\text{FD}_{\text{openl3}} \downarrow$ |
| DiT-SAO / v-pred | SAO-VAE | 0.32 | 0.63 | 134.73 | 0.25 | 2.94 | 104.35 |
| | + LAFA | **0.34** | **0.55** | **102.24** | **0.29** | **2.64** | **94.38** |
| DiT-SAO / RF | SAO-VAE | 0.35 | 0.59 | 122.78 | 0.22 | 2.84 | 94.35 |
| | + LAFA | 0.35 | 0.59 | **92.26** | **0.26** | **2.58** | **83.08** |
| DiT-ETTA / v-pred | SAO-VAE | 0.34 | 0.72 | 149.26 | 0.17 | 2.52 | 110.25 |
| | + LAFA | 0.34 | **0.63** | **140.35** | 0.17 | **2.20** | **93.93** |

**Training Objective** We use the standard rectified-flow loss

$$\mathcal{L}_{\text{RF}} = \left\| \hat{U} - (\varepsilon - Z_0) \right\|_2^2, \quad \hat{U} = \text{Dec}(Z_t; h, t).$$

By predicting masked channels in latent space, LAFA pushes the encoder toward higher-level, semantically coherent structure, while delegating low-level detail reconstruction to the frozen VAE decoder. This division of labor preserves locality and spectral regularity in the latent geometry, which we find beneficial for downstream diffusion training and convergence.

## 5 EXPERIMENTS

**Data** We train all models on open-source audio from FreeSound (FSD) and FMA. Following the Creative-Commons (CC) filtering protocol used in SAO, we start from a list of 486,493 CC-licensed recordings and remove overlaps to ensure no copyrighted content enters training. This yields a corpus of 436,602 recordings (6,207 h): 6,353 recordings from FMA (444 h) and 430,249 from FSD (5,763 h), all under CC-0, CC-BY, or CC-Sampling+.

**Evaluation** We evaluate latent audio diffusion models with three complementary metrics: i) $\text{FD}_{\text{OpenL3}}$ (lower is better): a Fréchet distance computed on OpenL3 embeddings, assessing overall realism and distributional closeness to references. ii) $\text{KL}_{\text{PaSST}}$ (lower is better): KL divergence between PaSST tag posteriors of generated and reference audio, measuring semantic correspondence. iii) CLAP score (higher is better): text–audio similarity from a Contrastive Language–Audio Pre-training model, measuring prompt adherence. For sound generation, we evaluate on AudioCaps; for music generation, on Song Describer. For reconstruction, we report STFT error, mel-spectrogram error, and FD metrics computed on the same evaluation sets.

### 5.1 TRAINING DETAILS

**VAE bottleneck (LAFA)** We train on 30 sec stereo audio chunks; clips shorter than 30 sec are padded with a training-time mask to maintain efficiency. We adopt SAO as the base waveform VAE (widely used in recent open-source systems) and fine-tune our LAFA module for 600k steps with batch size 64 and learning rate 5e-5.

**Diffusion transformers (DiTs)** All DiTs are trained for 1M steps. For ablation studies, we train each model for 250k steps unless otherwise stated. i) SAO-style DiT (v-objective): 24 layers, 24 heads, width 1536. Conditioning signals include text (T5-base encoder), timing (for variable-length synthesis), and diffusion timestep (sinusoidal embeddings). Conditioning is injected via cross-attention (text, timing) and/or prepended tokens (timing, timestep). ii) ETTA-style DiT: adopts AdaLN timestep conditioning; zero-initializes the final projection to match the VAE latent mean; uses GELU (tanh approximation) and rotary position embeddings (RoPE) with base 16,384.

### 5.2 BOOSTING DIT CONVERGENCE

We evaluate LAFA as a drop-in enhancement to an existing waveform VAE by comparing SAO-VAE with and without LAFA, across both diffusion objectives and architectures. As shown in Table 1,

Table 2: Generation performance on AudioCaps (higher ↑ / lower ↓ is better). † Contains AudioCaps in training data. *Fine-tuned on AudioCaps.

| Stage II | Stage I | Channels/sr | CLAP$_{score}$ ↑ | KL$_{PaSST}$ ↓ | FD$_{openl3}$ ↓ |
|---|---|---|---|---|---|
| Ground truth | – | – | 0.50 | – | – |
| AudioLDM2 † | MelVAE | 2/16kHz | **0.41** | **1.76** | 178.53 |
| TANGO2 † | MelVAE | 1/16kHz | 0.45 | 1.09 | 189.15 |
| SAO | SAO-VAE | 2/44.1kHz | 0.28 | 2.25 | **82.65** |
| DiT-SAO | SAO-VAE + LAFA | 2/44.1kHz | 0.26 | 2.58 | 83.08 |
| DiT-SAO-FT-AC* | SAO-VAE + LAFA | 2/44.1kHz | **0.41** | **1.76** | 64.72 |

Table 3: Generation performance on SongDescriber. We use MusicGen-large and AudioLDM2-music as the comparison open-source baselines. * denotes training on in-house data.

| Stage II | Stage I | Channels/sr | CLAP$_{score}$ ↑ | KL$_{PaSST}$ ↓ | FD$_{openl3}$ ↓ |
|---|---|---|---|---|---|
| Ground truth | – | – | 0.36 | – | – |
| MusicGen* | Encodec | 1/32kHz | 0.30 | 0.51 | 178.70 |
| AudioLDM2 | MelVAE | 2/16kHz | 0.31 | 0.66 | 286.24 |
| SAO | SAO-VAE | 2/44.1kHz | 0.36 | 0.61 | **119.53** |
| DiT-SAO | SAO-VAE + LAFA | 2/44.1kHz | **0.35** | **0.59** | 79.32 |

LAFA improves generation performance under both v-prediction and rectified flow, with rectified flow further offering superior training efficiency on audio and music generation. For diffusion architectures, LAFA consistently accelerates DiT convergence. Although DiT-ETTA is strong in its original setting, our experiments—trained on 430k samples versus over 1M in the original work—show that DiT-ETTA does not surpass DiT-SAO under limited data.

We also compare LAFA against state-of-the-art baselines. On AudioCaps (Table 2), training-data mismatch (Appendix B) prevents our DiT-SAO baseline from matching the official checkpoint, but LAFA still improves generation quality and matches SOTA FD. After finetuning on AudioCaps, LAFA yields substantial gains in CLAP and KL, outperforming AudioLDM2 under matched data. In music generation (Table 3), LAFA-augmented latents achieve better FD than open-source baselines, indicating that LAFA unlocks additional latent capacity. With more training data, CLAP scores will further improve (Appendix E), consistent with the scaling of text–audio alignment.

### 5.3 BALANCING INDUCTIVE BIAS AND HIGH COMPRESSION

To verify that LAFA does not degrade reconstruction quality, we evaluate both sound and music (Table 4) by comparing ground-truth and reconstructed audio using standard metrics: STFT distance, Mel distance, and SI-SDR (via the auraloss library (Steinmetz & Reiss, 2020), with default settings). The results show that the flow latent decoder predicts clean latents with reconstruction quality comparable to the original SAO-VAE across datasets. LAFA can introduce slight temporal misalignment relative to the ground truth, which mildly increases alignment-sensitive metrics such as STFT distance. However, perceptual evaluations in Appendix F confirm that LAFA does not compromise perceived reconstruction quality.

### 5.4 ABLATION STUDIES

Our goal is to introduce Mel-like spectral bias in the VAE latent space. In order to clarify the contribution of every parts in LAFA, we conduct ablation studies as follows.

**Component-wise Ablation of Mask Strategy.** We perform a component-wise ablation to assess the effect of LAFA's two masking designs: a causal mask, which imposes a monotonic channel ordering so that earlier channels predict later ones, and a span mask, which acts as a low-pass filter along this sequence. Using SongDescriber generation results at 50k training steps (where the DiT has converged), Table 5 shows that both masks are crucial. Using only the causal mask leads to

Table 4: Reconstruction performance on SongDescriber and AudioCaps. Lower is better for all metrics. $Mel_{dis}$ and $STFT_{dis}$ denote Mel and STFT distances, respectively.

| Model | Sampling rate | Frame rate | Channel size | SongDescriber (Music) | | AudioCaps (Sound) | |
|---|---|---|---|---|---|---|---|
| | | | | $Mel_{dis} \downarrow$ | $STFT_{dis} \downarrow$ | $Mel_{dis} \downarrow$ | $STFT_{dis} \downarrow$ |
| AudioLDM | 16kHz | 25.0Hz | $8 \times 16$ | 0.97 | 1.43 | 1.49 | **1.31** |
| SAO-VAE | 44.1kHz | 21.5Hz | 64 | **0.77** | 1.29 | **0.75** | **0.86** |
| + LAFA | 44.1kHz | 21.5Hz | 64 | **0.82** | **1.26** | 0.98 | 1.37 |

Table 5: Ablation Study in Mask Strategy. Results shows that both causal and span mask contributes to the LAFA benefits, removing either of them causes a significant generation quality drop.

| Causal Mask | Span Mask | $CLAP_{score} \uparrow$ | $KL_{PaSST} \downarrow$ | $FD_{openl3} \downarrow$ |
|---|---|---|---|---|
| ✓ | | 0.28 | 0.68 | 176.04 |
| | ✓ | 0.29 | 0.64 | 119.42 |
| ✓ | ✓ | **0.30** | **0.63** | **95.38** |

a skewed latent space in which earlier channels carry disproportionately more information, which is misaligned with the channel-agnostic nature of diffusion. Using only the span mask, without any ordering constraint on channels, weakens local correlations and hampers inpainting and encoder–decoder training, despite slightly smoothing the latent space. Combining both masks yields clear improvements, reducing FD from 168.69 to 95.38, indicating that their joint use is necessary.

**Design Choice of Latent Decoder.** To predict the clean latent after masking, we employ a rectified-flow latent decoder, which iteratively refines the latent variables and reduces the training-inference gap. To assess its contribution, we replace the rectified-flow decoder with a standard ViT decoder trained with an MSE loss on the masked latents. Even in this setting, the ViT encoder produces latents that outperform the vanilla VAE baseline, improving FD from 168.69 to 126.27 and indicating that the masking strategy alone benefits the latent geometry. However, under the same masking scheme, using the rectified-flow denoising decoder further reduces FD to 114.50. This suggests that the flow-based decoder further purifies the latent distribution, aligns it more closely with the VAE latent manifold, and enhances audio fidelity.

**Hyperparameter Study.** The masking ratio $p$ is the key hyperparameter in our masking strategy, as it determines the maximum span length $L_{mask} = p \times$ channel dim. At each step, we uniformly sample a mask length from $[0, L_{mask}]$ and apply span masks along the channel axis. Our ablations on the masking ratio show that: (1) at $p = 0.3$, LAFA does not yet exhibit a clear advantage than VAE baseline in audio generation, but the generation quality consistently improves as $p$ increases; (2) the best performance is achieved at $p = 1.0$. This aligns with our theoretical analysis in Section 4.2, where larger masked spans impose a stronger penalty on high-frequency components, yielding smoother latents that are better suited for diffusion modeling.

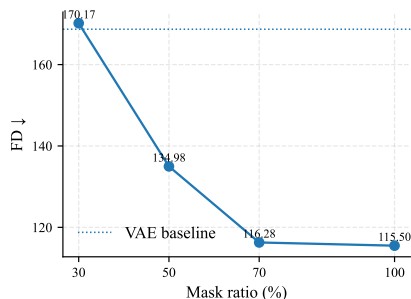

Figure 5: Masking ratio. Generation quality increases monotonically with the masking ratio. The y-axis reports FD on SongDescriber.

**Flow Decoder Inference Steps.** We employ a flow-based latent decoder to predict clean latents, whose iterative sampling alleviates the train–inference mismatch of conventional VAE decoders. To study the effect of the number of inference steps in the rectified flow decoder on generation quality, we vary the number of steps and report the results in Table 7. As the number of inference steps increases, the FD consistently decreases from 102.24 to 96.57, while CLAP and KL remain nearly unchanged. This indicates that the denoising decoder enhances overall perceptual audio quality without altering the semantic content of the generations. In practice, we adopt 2 steps for all generation experiments in this paper, as it provides a favorable trade-off between inference efficiency and generation quality.

Table 6: Ablation Study on Latent Decoder. Starting from the SAO-VAE baseline, we replace the LAFA latent decoder with a ViT decoder and compare both variants against LAFA. The results show that the masking strategy alone refines the latent space, while the denoise decoder further enhances generation quality through iterative sampling.

| Model | CLAP$_{score}$ ↑ | KL$_{PaSST}$ ↓ | FD$_{openl3}$ ↓ |
|---|---|---|---|
| baseline(SAO-VAE) | 0.27 | 0.79 | 168.69 |
| +LAFA w/ ViT Decoder | **0.33** | 0.71 | 126.27 |
| +LAFA w/ Denoise Decoder | 0.29 | **0.69** | **114.50** |

Table 7: Inference steps of the Flow Latent Decoder. The largest improvement in reconstruction quality occurs when increasing the number of steps from 1 to 2, while the generation quality continues to improve as the number of inference steps increases.

| Steps | MSE↓ | Mel$_{dis}$↓ | CLAP$_{score}$ ↑ | KL$_{PaSST}$ ↓ | FD$_{openl3}$ ↓ |
|---|---|---|---|---|---|
| 1 | 0.29 | 1.75 | 0.34 | 0.55 | 102.24 |
| 2 | 0.31 | 1.64 | 0.35 | 0.55 | 101.54 |
| 10 | 0.41 | 1.58 | 0.35 | 0.57 | 98.84 |
| 100 | 0.49 | 1.60 | 0.34 | 0.55 | 96.57 |

**Experiments on latent dimensionality.** To assess the generalizability of LAFA, we train SAO-VAEs with 64 and 128 dimensional latents. Although the 128-dim VAE is expected to yield better generation quality due to its higher reconstruction fidelity(see Appendix D), its latent space exhibits stronger high-frequency components (as observed in Figure 2), which imposes a heavier modeling burden on the DiT; consequently, its FD deteriorates from 175.57 in 64-dim to 313.68. Applying LAFA to both models, we observe that (1) LAFA provides consistent gains over the vanilla VAE for both 64 and 128 dimensions—particularly for the 128-dim case, where FD is reduced from 313.68 to 207.57, a 34% improvement, indicating that LAFA suppresses high-frequency components and makes the latent space more amenable to diffusion modeling; and (2) the 128-dim + LAFA model still does not surpass the 64-dim + LAFA variant. We attribute this to the increased complexity of modeling a larger latent space under the current LAFA capacity. Scaling up LAFA (Esser et al., 2024) is a promising direction for future work.

Table 8: Generalizability across latent dimensionality. 64d/128d denote VAE latent dimensionalities of 64 and 128, respectively. LAFA yields consistent improvements at both latent sizes, indicating that its benefits hold across different continuous-latent configurations.

| Model | CLAP$_{score}$ ↑ | KL$_{PaSST}$ ↓ | FD$_{openl3}$ ↓ |
|---|---|---|---|
| SAO-VAE-64d | 0.21 | 1.02 | 175.57 |
| SAO-VAE-64d + LAFA | 0.24 | 0.90 | 154.95 |
| SAO-VAE-128d | 0.18 | 1.12 | 313.68 |
| SAO-VAE-128d + LAFA | 0.20 | 0.85 | 207.57 |

## 6 CONCLUSION

This paper addresses the trade-off between the efficiency of waveform VAEs and the beneficial power-law spectral bias of Mel-spectrograms for audio diffusion. We show that VAE latents lack channel-wise locality, injecting excess high-frequency energy that conflicts with diffusion's coarse-to-fine dynamics. To address this, we introduce LAFA, a masked VAE bottleneck that reshapes latents via channel-span masking, effectively acting as a low-pass filter in the channel-frequency domain. LAFA accelerates Diffusion Transformer convergence and attains state-of-the-art or competitive audio and music generation performance without sacrificing reconstruction fidelity.

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

## A    DECLARATION OF LLM USAGE

We used large language models (LLMs) only as general-purpose assistive tools for grammar pol-ishing, minor rephrasing, and LaTeX formatting suggestions. All technical content ideas, methods, theory, experiments, hyperparameters, and analysis—was written and verified by the authors. We are fully responsible for the paper.

## B    MUSIC TRAIN DATA ORGANIZATION AND REPRODUCIBILITY

We construct natural-language prompts for text-to-music training from FMA metadata. SAO doesn't release their captions of training data, but only give the organization strategy for the original music metas, we follow their guide to construct the captions, which presented in the Appendix B, but maybe a concern for the mismatched performance.

Following Stable Audio Open, we sample a random subset of the fields year, genre, album, title, and artist, concatenate them into a prompt, shuffle the field order, and randomly vary casing. For half of the examples we retain field labels and join with commas; for the other half we join the values only. Unlike Stable Audio Open, we always keep the genre field, as it most directly characterizes musical content.

## C    ESTIMATING SPECTRAL DECAY IN LATENT SPACE

Specifically, we conduct spectral analysis as follows: Let $T$ and $C$ denote the numbers of time steps and channels, respectively, and let $X \in \mathbb{R}^{T \times C}$. Denote by $\widehat{X}[k_t, k_c]$ the 2-D discrete Fourier transform (DFT) of $X$ with the DC component at $(0, 0)$ and frequency indices

$$k_t \in \{-s\lfloor T/2 \rfloor, \dots, \lceil T/2 \rceil - 1\}, \qquad k_c \in \{-\lfloor C/2 \rfloor, \dots, \lceil C/2 \rceil - 1\}.$$

Define the 2-D power spectrum $P(k_t, k_c) = \left| \widehat{X}[k_t, k_c] \right|^2$.

We use the radially averaged power spectrum (RAPSD) as implemented in *pysteps* library[1], follow-ing the method of Ruzanski & Chandrasekar (2011). Let $l = \max(T, C)$ and, for each integer radius $r \in \{0, 1, \dots, \lfloor l/2 \rfloor\}$, define the ring

$$\mathcal{R}_r = \left\{ (k_t, k_c) : \text{round}\left( \sqrt{k_t^2 + k_c^2} \right) = r \right\}.$$

We associate each ring with a normalized radial frequency

$$f = \frac{r}{l} \in \left[ 0, \tfrac{1}{2} \right],$$

and define the one-dimensional (radially averaged) spectrum by

$$S(f) \equiv \frac{1}{|\mathcal{R}_r|} \sum_{(k_t, k_c) \in \mathcal{R}_r} P(k_t, k_c).$$

Given the radially frequency $S(f)$, we discard the DC bin and work with log–frequency coordinates $\ell = \log f$. After interpolating $S(f)$ onto an equally spaced log grid $\{\tilde{\ell}_j\}_{j=1}^{J}$, we fit

$$\log S(\tilde{\ell}_j) \approx -\alpha \tilde{\ell}_j + b$$

by least squares. The slope $\alpha$ estimates the spectral decay exponent, with

$$S(f) \propto f^{-\alpha}$$

---

[1]https://github.com/pySTEPS/pysteps

## D    RECONSTRUCTION OF DIFFERENT VAEs

We train SAO-VAEs with different latent dimensionalities and evaluate their reconstruction performance on SongDescriber, as in Table 9. As expected, increasing the latent capacity improves reconstruction fidelity, with both Mel distance and STFT distance decreasing monotonically. However, although one might anticipate the best generation performance at 128 dimensions, this is not the case (Section 8). Higher-dimensional latents introduce stronger high-frequency components to preserve fine-grained audio details (Figure 2) and reduce locality in the latent space (Figure 3), which increases the burden on the generative model. LAFA is specifically designed to mitigate this reconstruction–generation trade-off.

Table 9: Reconstruction of different SAO-VAEs on SongDescriber.

| Model | $Mel_{dis} \downarrow$ | $STFT_{dis}$ |
|---|---|---|
| SAO-VAE-32d | 0.93 | 1.22 |
| SAO-VAE-64d | 0.75 | 1.09 |
| SAO-VAE-128d | **0.59** | **0.94** |

## E    INFLUENCE OF DATA ON CLAP PERFORMANCE

We evaluate generation performance under a fixed prompt construction strategy while varying the amount of training data, as in Table10. As the dataset size increases from 90k to 430k examples, the CLAP score improves from 0.31 to 0.35. These results indicate that CLAP performance depends not only on caption quality but also on the volume of training data.

Table 10: CLAP scores on SongDescriber with varying amounts of training data. The scores improve consistently as the pretraining corpus is scaled up.

| Training data | $CLAP_{score} \uparrow$ |
|---|---|
| 90k | 0.31 |
| 430k | 0.35 |

## F    MORE RECONSTRUCTION PERFORMANCE OF LAFA

LAFA may exhibit slight temporal misalignment with the ground-truth audio due to the rectified-flow latent decoder: the predicted latent can reconstruct audio that is perceptually similar but not sample-aligned, which leads to higher errors on temporal alignment metrics such as STFT distance. However, when evaluated with perceptual metrics such as Tjandra et al. (2025), LAFA does not compromise reconstruction quality. Its perceptual fidelity matches that of the vanilla WaveVAE and significantly surpasses MelVAE.

Table 11: Perceptual Quality of VAEs on AudioCaps.

| Model | CE $\uparrow$ | CU $\uparrow$ | PC $\uparrow$ | PQ $\uparrow$ |
|---|---|---|---|---|
| AudioLDM | 3.38 | **4.89** | 3.28 | 5.52 |
| SAO-VAE | 3.47 | 4.84 | **3.50** | 5.68 |
| + LAFA | **3.48** | 4.86 | 3.43 | **5.70** |

## G    VISUALIZATION OF VAE CHANNELS

To assess whether information is evenly distributed across latent channels—or whether some channels are crucial while others are redundant, we perform a residual visualization experiment. Given a latent representation $z \in \mathbb{R}^{B \times C \times T}$, we systematically ablate each channel and examine its effect on reconstruction. Specifically, for each channel $c \in \{1, \ldots, C\}$, we replace the corresponding latent activations with Gaussian noise of small variance, while keeping the remaining channels unchanged. The perturbed latent code is then decoded back into waveform space.

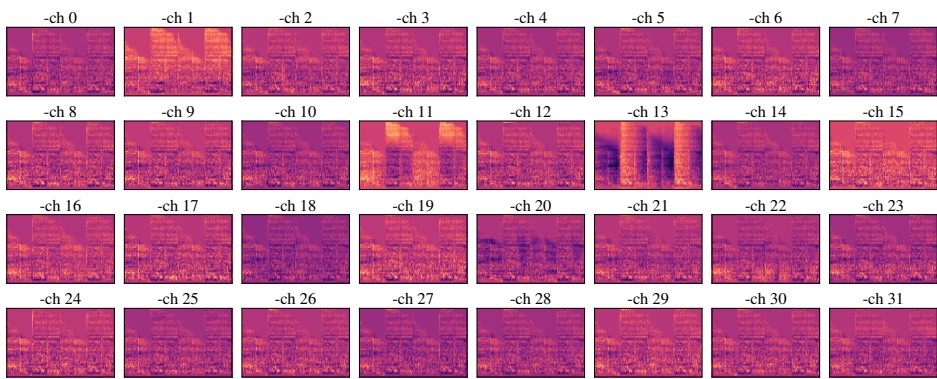

Figure 6: Residual Information of each channel in 32dim VAE

From the reconstructed waveform, we compute the difference with respect to the ground-truth signal in the mel-spectrogram domain. These residual mel-spectrograms capture the contribution of each channel to the reconstruction. By arranging the residuals across all channels in a grid, we obtain a visual map of channel-wise effects, where stronger and more structured residuals indicate channels encoding salient information, whereas weak or noise-like residuals suggest redundant or noisy channels.

