# OpenReview forum: "Listens like Mel: Boosting Latent Audio Diffusion with Channel Locality"
_ICLR.cc/2026/Conference — Submitted to ICLR 2026_

### Official Review · Reviewer_Xq35 · 2025-10-30

**Soundness:** 2
**Presentation:** 1
**Contribution:** 2
**Rating:** 2
**Confidence:** 3

**Summary:**

This paper shows that the latents of waveform VAEs do not well align with the Mel-spectrogram's approximate power-law spectrum, making it not suitable for training diffusion models. Authors proposed a channel-span masking bottleneck (LAFA) to improve the VAE latents, hence speeding up the convergence of diffusion-based audio generation models.

Authors conducted experiments on audio and music generation tasks to show the effectiveness of LAFA.

**Strengths:**

- An intuitive and lightweight method of speeding up the convergence of diffusion models for audio generation.

**Weaknesses:**

- The paper is difficult to follow, the presentation and writing need to be largely improved.

- Potential over-claims. Authors claimed that 4x fast convergence is achieved using LAFA, but no evidences provided. According to Figure  1, LAFA is not 4x faster in terms of the number of training iterations and FD performance.

- The improvements brought by LAFA seems not consistent across the evaluation tasks. LAFA mostly introduces consistent improvements on one metric (FD_openl3), while the other results are mixing (Tab1 and Tab2). This is also true when looking at the reconstruction  performance (Table 4): SAO-VAE+LAFA performs worse than SAO-VAE in general.

**Questions:**

- Table 1: I feel it is unfair to compare DiT-SAO-FC-AC with other baselines like SAO, which are not finetuned on AudioCaps. Am I misunderstanding something?

---

> ### Author Response · Authors · 2025-12-03
> **Response to Reviewer Xq35**
>
> We appreciate the reviewer's suggestions and comments. We will resubmit a paper with more ablation experiments and corrected writing issues.
>
> ---
>
> # Writing and presentation
> We acknowledge that the initial submission was difficult to follow in places, especially around the method. In the revised manuscript we will:
>
> - streamline the exposition of spectral analysis and locality metrics;
> - clarify the LAFA architecture with an improved version and more precise notation
>
> We appreciate that other reviewers found the core idea understandable, but we agree the paper can be written more clearly and will revise accordingly.
>
> # Response to the question about Figure 1
>
> Thank you for the question. We have added auxiliary dashed lines to make Figure 1 easier to read in our revised manuscript. In our experiments, LAFA accelerates DiT convergence by 4× in the DiT-ETTA setting and by 2.5× in the DiT-SAO setting. We will update the main text to say “up to 4× faster convergence” and clearly tie this to the DiT-ETTA result.
>
> # Response to the experiment questions
>
> ## Why does LAFA appear to give worse audio generation performance in Table 1?
>
> Here is a misunderstanding in Table 1, and we appreciate the opportunity to clarify it.
>
> In Table 1 (and Table 2), we report results from the official SAO checkpoint for a fair comparison with prior work. However, our own reimplementation of SAO—used in **Table 3** for ablations—is weaker than the official checkpoint. This discrepancy arises from two factors:
>
> - Lack of original training data. SAO doesn't release their captions of training data, but only give the organization strategy for the original music metas, we follow their guide to construct the captions, which presented in the Appendix B, but maybe a concern for the mismatched performance.
> - Misalignment of training resources. Due to limited resources, we trained generation DiT with **30s** sequence length, while official SAO trained with **47s** sequence length.
>
> And we found that LAFA is better than the SAO baseline under the same training resource consumption, which is also mentioned in Table 3.
>
> ## Why does LAFA slightly degrade CLAP in Table 2?
> The small drop in CLAP is largely due to the caption differences between the training data, as discussed earlier. Despite this, LAFA consistently improves KL and FD, indicating stronger semantic consistency in the generated audio and higher overall audio quality.
>
> ## Reconstruction performance of LAFA
> LAFA may exhibit slight temporal misalignment with the ground-truth audio due to the rectified-flow latent decoder: the predicted latent can reconstruct audio that is perceptually similar but **not sample-aligned**, which leads to higher errors on temporal alignment metrics such as STFT distance.
>
> |Model|CE&uarr;|CU&uarr;|PC&uarr;|PQ&uarr;|
> |---|---|---|---|---|
> |AudioLDM|3.38|**4.89**|3.28|5.52|
> |SAO-VAE|3.47|4.84|**3.50**|5.68|
> |+LAFA|**3.48**|4.86|3.43|**5.70**|
>
> However, when evaluated with **perceptual metrics** such as AudioBox[1], which also used in [2], LAFA does not compromise reconstruction quality. Its perceptual fidelity matches that of the vanilla WaveVAE and significantly surpasses MelVAE.
>
> This is consistent with listening tests on our reconstruction demos (https://iclr-lafa.github.io/lafa.github.io/), where LAFA produces audio that is perceptually indistinguishable from SAO-VAE. We will include the AudioBox reconstruction results in the extended version of the paper.
>
>
> # Response to the question in Table 1
>
> We appreciate your concern regarding fair comparison, and we apologize for any confusion caused.
>
> We include the **DiT-SAO-FT-AC** setting for the following reason: both AudioLDM2 and Tango2 incorporate AudioCaps in their training data. Since our baseline SAO and LAFA models do not use AudioCaps during pretraining, a direct comparison would be unfair. Finetuning on AudioCaps allows us to control for this data mismatch.
>
> Even without AudioCaps, WaveVAE achieves better FD than MelVAE, confirming its advantage in audio fidelity. After finetuning on AudioCaps, LAFA shows substantial improvements in CLAP and KL and outperforms AudioLDM2, demonstrating the potential of the LAFA architecture under **matched data** conditions.
>
> We will add these details to the table caption to make the comparison clearer. Thank you for raising this important point.
>
> ---
>
> ## References
>
> [1] Andros Tjandra, Yi-Chiao Wu, Baishan Guo, John Hoffman, Brian Ellis, Apoorv Vyas, Bowen Shi, Sanyuan Chen, Matt Le, Nick Zacharov, et al. Meta audiobox aesthetics: Unified automatic quality assessment for speech, music, and sound. arXiv preprint arXiv:2502.05139, 2025.
>
> [2] Yushen Chen, Kai Hu, Long Zhou, Shulin Feng, Xusheng Yang, Hangting Chen, and Xie Chen. Auv: Teaching audio universal vector quantization with single nested codebook. arXiv preprint arXiv:2509.21968, 2025.

---

### Official Review · Reviewer_awW2 · 2025-10-31

**Soundness:** 3
**Presentation:** 3
**Contribution:** 2
**Rating:** 4
**Confidence:** 3

**Summary:**

his paper identifies that VAE latent representations, unlike Mel-spectrograms, contain excessive high-frequency energy along the channel axis, which is detrimental to diffusion model training. To resolve this, it introduces Latent Flow MAE (LAFA), a plug-in module that applies channel-span masking to the VAE latent space. The authors theorize this acts as a low-pass filter, inducing a more favorable, Mel-like spectral bias. Experiments show LAFA accelerates Diffusion Transformer convergence on audio generation tasks and improves final performance without sacrificing reconstruction fidelity.

**Strengths:**

- LAFA is an intuitive and simple plug-in module that demonstrably leads to faster convergence and better generation quality. Its modularity is a significant practical advantage.

- The paper provides a theoretical justification for its method, framing channel masking as a low-pass filter. This offers a helpful intuition for why the approach is effective.

**Weaknesses:**

- While quantitative metrics improve over the baseline, the practical significance of these gains is questionable. The qualitative results do not demonstrate a clear perceptual advantage over existing open-source models like MusicGen. Furthermore, the output quality lags significantly behind commercial state-of-the-art models (e.g., Suno), making the overall contribution feel incremental.

- All experiments are performed on a single VAE architecture (SAO-VAE). Without tests on other diverse codecs (e.g., Mel-based VAEs, Encodec), the claim of general applicability is not fully substantiated.

-  The paper lacks crucial ablations. It does not empirically compare the chosen span masking against simpler uniform masking, nor does it analyze the sensitivity of the model to the mask ratio, a key hyperparameter.

**Questions:**

- Have you tested LAFA on VAE architectures other than SAO-VAE, such as the one from AudioLDM or an Encodec-style model, to verify its generalizability?
- Could you provide an empirical comparison between contiguous span masking and random uniform masking of channels to justify your design choice?
- How does model performance vary with different mask ratios? Is there an optimal range for this hyperparameter?

---

> ### Author Response · Authors · 2025-12-02
> **Response to Reviewer awW2**
>
> We appreciate the reviewer's suggestions and comments. We will resubmit a paper with more ablation experiments and corrected writing issues.
>
> ---
>
> # Quantitative metrics in Table 2
> > The qualitative results do not demonstrate a clear perceptual advantage over existing open-source models like MusicGen. Furthermore, the output quality lags significantly behind commercial state-of-the-art models (e.g., Suno)
>
> For a fair comparison, we benchmark against MusicGen, the strongest open-source model specialized for music generation. LAFA achieves higher CLAP and FD scores than MusicGen, although KL is slightly lower. This gap is largely attributable to differences in training data scale: MusicGen is trained on ~20,000 hours of in-house music, whereas LAFA is trained on only ~3,000 hours of public-domain music and sound, with copyrighted material filtered out. The disparity is even more pronounced when comparing to commercial systems such as Suno, whose data scale and content remain undisclosed.
>
>
> # Generalizability beyond SAO-VAE
> > All experiments are performed on a single VAE architecture (SAO-VAE). Without tests on other diverse codecs (e.g., Mel-based VAEs, Encodec), the claim of general applicability is not fully substantiated.
>
> Our focus is on continuous WaveVAE-style representations, which preserve lossless, high-fidelity reconstruction at 44.1 kHz. The goal of LAFA is to combine the strong reconstruction of WaveVAEs with the generation-friendly smoothness of Mel-like representations.
>
> Mel-based VAEs and Encodec-style models are lossy and therefore unsuitable for our objectives, as their reduced reconstruction fidelity limits downstream audio quality. Because SAO-VAE is the most widely adopted high-fidelity VAE in recent works (e.g., ETTA[1], DiffRhythm[2], EZAudio[3]), we use it as our primary backbone.
>
> |VAEs | CLAP&uarr; | KL &darr; | FD &darr; |
> |---|---|---|---|
> |SAO-64d|0.21|1.02|175.57|
> |SAO-64d + LAFA | **0.24**|0.90|**154.95**|
> |SAO-128d|0.18|1.12|313.68|
> SAO-128d+LAFA|0.20|**0.85**|207.57|
>
> To assess generality within this family, we evaluate LAFA on 64dim and 128dim variants of SAO-VAE, and observe consistent improvements across both latent sizes. This demonstrates that LAFA benefits hold across different continuous-latent configurations.
>
> # Additional ablation studies
> > The paper lacks crucial ablations. It does not empirically compare the chosen span masking against simpler uniform masking, nor does it analyze the sensitivity of the model to the mask ratio, a key hyperparameter.
>
> We have added the experiments of the mask ratio. Our span masking is randomly processed each step: that we have mask_ratio to controls the highest mask length $L_{mask}$ = mask_ratio $\times$ channel_dim, the total possible masked length is bounded by $L_{mask}$. At each training step, we uniformly sample a mask length from $[0, L_{mask}]$ and divide it into 1-10 contiguous spans.
>
> | Mask ratio | CLAP ↑|        KL↓ |        FD↓    |
> |----|----|----|----|
> | 0.3 |        0.23|        0.82|        170.17|
> | 0.5 |        0.26|        0.80|        134.98|
> | 0.7 |        0.28|        0.70|        116.28|
> | 1.0 |        **0.29**|        **0.69**|        **115.50**|
>
> Our findings:
> - Reducing the mask ratio makes span masking resemble uniform random masking, leading to worse generation performance.
> The inpainting task becomes too trivial, and the encoder primarily copies local information rather than reshaping the latent geometry.
> - Increasing the mask ratio strengthens the frequency-selective penalty predicted by our theory (Sec. 4.2), accelerating convergence and improving both KL and FD.
> - The best performance is achieved at mask ratio = 1.0.
>
> ---
> # Response to questions
> > Have you tested LAFA on VAE architectures other than SAO-VAE, such as the one from AudioLDM or an Encodec-style model, to verify its generalizability?
>
> Our aim is to enhance continuous WaveVAE representations that support high-fidelity reconstruction. Mel-VAEs (AudioLDM) and Encodec-style models are lossy and therefore inconsistent with this objective.
>
> SAO-VAE is the most popular VAE around audio generation, which is used by ETTA[1], Diffrhythm[2], EZAudio[3]. So we choose SAO-VAE as our base model. Moreover, we conduct LAFA experiments on dim128 VAE, which also brings promotion in audio generation, as discussed in **Generalizability beyond SAO-VAE**.
>
> > Could you provide an empirical comparison between contiguous span masking and random uniform masking of channels to justify your design choice?
>
> Provided in **Additional ablation studies**: decreasing mask ratio causes span masking to degenerate toward uniform masking, which leads to degraded performance.
>
> > How does model performance vary with different mask ratios? Is there an optimal range for this hyperparameter?
>
> Also addressed in **Additional ablation studies**: model performance improves monotonically with mask ratio, with the optimal range near 1.0.

---

> ### Author Response · Authors · 2025-12-02
> **Response to Reviewer awW2 (2)**
>
> ## References
> [1] Sang-gil Lee, Zhifeng Kong, Arushi Goel, Sungwon Kim, Rafael Valle, and Bryan Catanzaro. Etta: Elucidating the design space of text-to-audio models. arXiv preprint arXiv:2412.19351, 2024.
>
> [2] Yuepeng Jiang, Huakang Chen, Ziqian Ning, Jixun Yao, Zerui Han, Di Wu, Meng Meng, Jian Luan, Zhonghua Fu, and Lei Xie. Diffrhythm 2: Efficient and high fidelity song generation via block flow matching. arXiv preprint arXiv:2510.22950, 2025.
>
> [3] Jiarui Hai, Yong Xu, Hao Zhang, Chenxing Li, Helin Wang, Mounya Elhilali, and Dong Yu. Ezaudio: Enhancing text-to-audio generation with efficient diffusion transformer. arXiv preprint arXiv:2409.10819, 2024.

---

### Official Review · Reviewer_hLGW · 2025-11-01

**Soundness:** 2
**Presentation:** 1
**Contribution:** 2
**Rating:** 4
**Confidence:** 4

**Summary:**

This paper focuses on the property of the latent space created by audio VAEs. The observation is that, if the latent space can be properly processed or regularized, the training of the audio generative model on this latent space will converge to a target performance faster or eventually results in better performance.
The proposed latent space processing/regularization method is called "LAFA".

The paper contains some interesting visualization of the latent spaces from different audio VAEs. Evaluations show that the proposed method can lead to SOTA or close-to-SOTA performance in either audio generation or music generation tasks.

Overall, the paper is interesting and presents meaningful insights to the community. However, as a reader, I do feel some missing experiments, over claiming, and some confusing paper writings.

I believe the paper is in the borderline for acceptance.

**Strengths:**

- Visualization of latent space properties
- Solid proposed method and evaluation results

**Weaknesses:**

### Minor issues
- Missing experimental results
    - Line 240: I cannot see the visualization of Mel features in Figure 3
    - Line 416: I cannot find related information in Appendix
- Wrong interpretation
    - Line 411. The authors say "LAFA consistently improves FDOpenL3 and KLPaSST for sound generation". However, if we check table.1, before AudioCaps finetuning, the proposed DiT-SAO has worse metrics compared the SAO
### Major issues
- Lack of ablation studies on the hyper-parameters of LAFA
    - LAFA masks out a part of the latent channels and then use Flow-Matching to recover the information loss.
    - While the final masking strategy is mentioned in the paper, we don't how this strategy is designed, we don't know how sensitive LAFA is to the hyper-parameter of masking
- Lack of ablation study on the masking idea itself
    - The proposed method masks out a part of the latent channels, and then process the masked features with a Transformer encoder and a Flow-matching decoder: it is a denoising auto-encoder for the latent features!
        - For a perfect denoising auto-encoder, the target is to reconstruct the clean input. In this paper, the decoded latent features exhibit different properties from the input, so it is not designed to be a perfect auto-encoder
            - Hereby we have a question: which one has contributed more to the get the improved latent features, the mask or the Flow-matching decoding?
            - We need ablation studies to anwser this question
                - Bypass flow-matching, simply use MSE to train the auto-encoder
                - Bypass masking, simple use the current encoder + Flow-matching decoder pipeline to process the latent features
                - Is the encoder really needed? No experiment to prove it.
- Missing of other experiments
    - Figure 3 reveals that the latent space of a VAE with 64dim + LAFA (proposed method) looks similar to dim32
        - Table.4 only compared the 64dim VAE with and without LAFA.
        - Why don't we compare dim32 with dim64 + LAFA?
    - Will dim128 makes the training of audio generation model more difficult? If so, why not present the results for dim128 + LAFA?
        - I am asking this, because audio generation models can be well trained on 64dim latent space without LAFA (though I agree that LAFA makes this easier). I want to know if LAFA can convert a very difficult case such as dim128 or dim256 into a trainable latent space.
        - Such experiments can emphasize the value of LAFA

**Questions:**

Please refer to my comments above

---

> ### Author Response · Authors · 2025-12-02
> **Response to Reviewer hLGW (1)**
>
> # Response to Minor issues
> We thank the reviewer for pointing out these omissions.
>
> > Line 240: I cannot see the visualization of Mel features in Figure 3
>
> We will restore this plot and ensure they are correctly referenced as Fig. 3.
>
> > Line 416: I cannot find related information in Appendix
> The missing appendix table reports the effect of data scaling on CLAP:
>
> |Training Data|CLAP &uarr;|
> |---|---|
> |90k|0.31|
> |430k|0.35|
>
> We compare generation performance under the same prompt construction strategy but with different amounts of training data. The results show that CLAP depends not only on caption quality, but also on training data size, and increases consistently as we scale the pretraining corpus. We will restore this table in the Appendix and properly reference it from the main text.
>
> ## Wrong interpretation (Line 411 / Table 1)
> > Line 411. The authors say "LAFA consistently improves FDOpenL3 and KLPaSST for sound generation". However, if we check table.1, before AudioCaps finetuning, the proposed DiT-SAO has worse metrics compared the SAO
>
> There is a misunderstanding in our original explanation of Table 1, and we appreciate the opportunity to clarify it. Our reimplementation of SAO underperforms the official SAO (in Table 3). However, we show the results of official SAO checkpoint in Table 1 and Table 2 for fair comparison.
> Our reimplementation of SAO underperforms the official SAO checkpoints due to:
> **Training data mismatch.** SAO does not release their original captions, only metadata and a captioning strategy. We reconstruct captions following their procedure (Appendix B), which likely introduces a performance gap.
>
> **Training-resource mismatch.** Our DiT is trained with 30 s audio segments, while official SAO uses 47 s.
>
> Under the same training budget and data (our SAO baseline vs. LAFA), LAFA consistently outperforms SAO, as also reflected in **Table 3**. We will revise the text around Line 411 to clarify that the “consistent improvement” claim refers to comparisons with our SAO baseline under matched conditions.
>
> # Response to Major issues
>
> ## Ablation on LAFA hyperparameters (mask ratio)
>
> The mask ratio is indeed the key hyperparameter in our masking strategy. It controls the maximum span length
> $L_{mask}$ = mask_ratio $\times$ channel_dim.
> At each step, we uniformly sample a mask length from $[0, L_{mask}]$ and apply span masks along the channel axis.
>
> | Mask ratio | CLAP &uarr;|KL&darr;|FD&darr;|
> |---|---|---|---|
> |0.3|	0.23|0.82 |170.17|
> |0.5|	0.26|0.80	|134.98|
> |0.7|	0.28|0.70	|116.28|
> |1.0|	**0.29**|**0.69**|**115.50**|
>
> Our mask-ratio ablations (now added) show that: 1) Generation quality improves as the mask ratio increases; 2) The best performance is obtained at mask ratio = 1.0.
>
> This is consistent with our theory in **Section 4.2**: larger masked spans increase the effective penalty on high-frequency components, producing smoother latents that are better suited for diffusion modeling. We will include these results in the revised manuscript.
>
> ## Ablation on the masking idea vs. flow-matching
>
> **Clarification of LAFA’s role.** First, we clarify a misunderstanding: although we mask part of the latent channels and decode them with a rectified-flow decoder, the downstream generation DiT uses the output of the ViT encoder, not the decoded latents (as stated at Line 353). LAFA is thus MAE-like: masking and denoising are used to enrich the bottleneck representation.
>
> We address the reviewer’s questions via ablations:
>
> |Latent Decoder | CLAP&uarr; | KL &darr; | FD &darr; |
> |---|---|---|---|
> | baseline | 0.27| 0.79| 168.69|
> | ViT decoder | **0.33** | 0.71 | 126.27 |
> |Denoise decoder | 0.29| **0.69** | **114.50**|
>
> **(1) Effect of masking (without flow).**
> We replace the rectified-flow decoder with a standard ViT decoder and train with an MSE loss on the masked latents. Even in this case, the ViT encoder yields latents that outperform the raw VAE baseline in generation metrics, indicating that the masking strategy alone improves the latent geometry.
>
> **(2) Effect of the denoising (flow) decoder.**
> Comparing ViT decoder vs. rectified-flow decoder under the same masking scheme, we observe that the ViT decoder sometimes achieves slightly better CLAP but consistently worse FD and KL. This suggests that the flow-based decoder further “purifies” the latent distribution, bringing it closer to the VAE decoding manifold and improving audio fidelity.
>
> **(3) Necessity of the encoder.**
> The newly trained ViT encoder defines a higher-level, semantically structured latent space, while the frozen VAE handles low-level waveform reconstruction. Removing the encoder would collapse LAFA back to the original VAE and eliminate the main benefit. In this sense, the encoder is essential to our design.
>
> We will summarize these ablations and clarify the roles of encoder vs. decoder in the revised version.

---

> ### Author Response · Authors · 2025-12-02
> **Response to Reviewer hLGW (2)**
>
> ## Additional experiments on latent dimensionality
> > Figure 3 reveals that the latent space of a VAE with 64dim + LAFA (proposed method) looks similar to dim32
> Table.4 only compared the 64dim VAE with and without LAFA.
> Why don't we compare dim32 with dim64 + LAFA?
>
> > Will dim128 makes the training of audio generation model more difficult? If so, why not present the results for dim128 + LAFA?
>
> We appreciate this suggestion; it aligns with our aim of alleviating the reconstruction–generation trade-off.
>
> |VAEs | CLAP&uarr; | KL &darr; | FD &darr; |
> |---|---|---|---|
> |SAO-32d| **0.26**|0.91|277.29|
> |SAO-64d|0.21|1.02|175.57|
> |SAO-64d + LAFA | 0.24|0.90|**154.95**|
> |SAO-128d|0.18|1.12|313.68|
> SAO-128d+LAFA|0.20|**0.85**|207.57|
>
> ### Experiments on 32dim VAE
> Although 32dim VAE latents are smoother and easier for generation models to learn, their reconstruction quality is poor. While the 32dim VAE achieves strong semantic metrics such as KL and CLAP, it consistently yields worse FD scores than the 64dim VAE, indicating lower audio fidelity. Thus, despite faster convergence, a 32dim latent space is not an attractive operating point.
>
> ### Experiments on 64dim and 128dim
> We train 64dim and 128dim SAO-VAEs, both with and without LAFA, and observe:
> - LAFA provides consistent gains over the vanilla VAE for both 64dim and 128dim.
> - 128dim + LAFA does not surpass 64dim + LAFA. We attribute this to the increased complexity of modeling a larger latent space with the current LAFA capacity. Scaling up LAFA (e.g., larger Transformer/flow backbones, as in Stable Diffusion 3[1]) is a promising direction, which we leave for future work.
>
> We will explicitly report the 32dim/64dim/128dim comparisons to emphasize that LAFA helps make higher-dimensional, high-fidelity VAEs more generation-friendly, while very low-dimensional VAEs suffer from unacceptable reconstruction loss.
>
> ## Reference
> [1] Patrick Esser, Sumith Kulal, Andreas Blattmann, Rahim Entezari, Jonas M¨uller, Harry Saini, Yam Levi, Dominik Lorenz, Axel Sauer, Frederic Boesel, et al. Scaling rectified flow transformers for high-resolution image synthesis. In Forty-first international conference on machine learning, 2024.

---

### Official Review · Reviewer_DYP6 · 2025-11-01

**Soundness:** 3
**Presentation:** 2
**Contribution:** 3
**Rating:** 6
**Confidence:** 2

**Summary:**

This paper makes the observation that audio spectrograms exhibit certain characteristics --- specifically a power law spectrum --- that do not get appropriately generated by diffusion models primarily because the latest space of VAEs do not accurately learn this characteristic. To fix this, the paper proposes a sequence of masking techniques which results in boosting the effect of channel locality, thereby better fitting the power law spectrum behavior. A side benefit is that this also speeds up the DiT transformer without compromising reconstruction and compression. Experimental results are good.

**Strengths:**

-- The observation that the latent space learnt by VAEs often exhibit amplified high frequency energy along the channel axis, thereby deviating from the Mel power-law bias, is certainly an interesting observation. If this observation holds, meaning that it is fundamental to any VAE and not an outcome of some hyper-parameter choice, then this is a fundamental observation and remedies to this (some of which are proposed by the paper) should be an important boost to audio generation performance.

-- The masking idea is intuitive in the sense that the VAE is forced to learn a specific time-frequency bucket from the neighboring buckets, forcing locality and smoothness. Causality can also be enabled by designing the masks appropriately.

-- Some parts of the paper are amazingly clear, while other parts are sort of the vice versa (and these parts seem interspersed).

-- The theoretical analysis is pointing to an interesting observation but I may not have fully understood the inpainting part of the argument. I need to think more here, however, I see that authors are trying to demonstrate that the masking indeed leads to a low-pass filtering effect, which justifies why LAFA is able to preserve more energy in the lower frequencies. Is that the main point here?

-- Experiment results are quite good in some settings.

**Weaknesses:**

-- A point that is bothering me is why is it that VAEs are not able to learn the MEL behavior. If one designs a loss function over a spectrogram and ensures adequate weights for the reconstruction loss, the VAE should learn to invest energy in the power-law pattern. I might be missing something here.

-- The paper could better motivate various design choices (or at least explain them better). All the descriptions around Fig 4 is a bit quick and unclear. Section 4.1 has many parts that are unclear.

-- The paper seems to have a lot of typos. Fig 3 is referred to, where the authors probably meant Fig 2. S(f) is not defined in L192. The flow of presentation can be much better.

-- The presentation of the experiment section is confusing. Between row 4 and 5, is the gain due to LAFA or due to FT-AC? If MelVAE is better, why is LAFA needed? CLAP score seems to degrade in Table 2. In several cases of Table 3, seems like the gain from LAFA is almost negligible.



-- I am trying to understand why the masking technique, in the space of design choices to address the problem, is the correct one. Or at least, why did authors decide to adopt this approach? I do see that masking is bringing the VAE output closer to the MEL behavior, so I

**Questions:**

See above.

---

> ### Author Response · Authors · 2025-12-02
> **Response to Reviewer DYP6**
>
> # Question about theoretical analysis
> > The theoretical analysis is pointing to an interesting observation but I may not have fully understood the inpainting part of the argument. I need to think more here, however, I see that authors are trying to demonstrate that the masking indeed leads to a low-pass filtering effect, which justifies why LAFA is able to preserve more energy in the lower frequencies. Is that the main point here?
>
> Yes, that is precisely the point of our analysis. Span masking, combined with an inpainting operator, induces an implicit low-pass prior on the latent representation, which LAFA leverages.
>
> Formally, we apply a span mask $M$ to the latent $z$, obtaining $Mz$, and inpaint the missing region $(I-M)z$ with a smoothing operator $S$ (a circulant kernel). The predicted latent is
>
> $A_M z = Mz + (I-M)Sz.$
>
> Analyzing the reconstruction objective
>
> $\mathbb{E}_M \|A_M z - z\|^2 = \sum_k \lambda_k |z_k|^2,\quad \lambda_k \approx p\|1 - H(k)|^2,$
>
> where $p = \text{(expected masked length)}/C$ is proportional to the span length. For a low‑pass S, we have $|H(k)|\approx 1$ at small $k$ and decreasing magnitude at higher $k$, so $|1-H(k)|^2$ is tiny at low frequencies and large at high frequencies.
>
>
> | $p$ | CLAP&uarr; | KL &darr; | FD &darr; |
> |---|---|---|---|
> | 0.3 | 0.23|0.82|170.17 |
> | 0.5 |0.26|0.80| 134.98 |
> | 0.7 |0.28|0.70| 116.28 |
> | 1.0 |**0.29**|**0.69**|**115.50**|
>
> As $p$ increases, this penalty is amplified, further suppressing high-frequency energy and producing smoother latents.
>
> # Weakness Discussion
>
> ## 1 Importance of Bottleneck Fine-Tuning
> > A point that is bothering me is why is it that VAEs are not able to learn the MEL behavior. If one designs a loss function over a spectrogram and ensures adequate weights for reconstruction loss, the VAE should learn to invest energy in the power-law pattern. I might be missing something here.
>
> Imposing a Mel bias via a Mel-spectrogram loss on the waveform reconstruction is indeed a natural idea, and has been widely adopted in previous vocoder and VAE works (e.g., BigVGAN2[1], DiffRhythm2[2]), where a Mel loss between the decoded and ground-truth waveform is combined with the reconstruction and KL terms. This objective, however, is primarily optimized for **reconstruction quality**: the Mel spectrogram is used because it correlates well with human perception.
>
> Crucially, such a reconstruction-oriented Mel loss does **not** directly constrain the structure of the latent space to follow the power-law pattern we target. In practice, the optimization can push the decoder to absorb most of the burden of matching the Mel features, while the latent representation itself remains suboptimal. This is why several recent tokenizers (e.g., FlexTok[3], ALMTokenizer[4]) freeze the autoencoder and explicitly shape the bottleneck with additional losses. LAFA follows this line of work.
>
> We will clarify this distinction in the revised version.
>
> ## 2 Design Choices in LAFA
> > The paper could better motivate various design choices (or at least explain them better). All the descriptions around Fig 4 is a bit quick and unclear. Section 4.1 has many parts that are unclear.
>
> Our design is motivated by efficiency and by the need to explicitly shape the latent distribution:
>
> **Base architecture.** LAFA adopts a MAE-style architecture with a Transformer encoder–decoder, but operates on VAE latents as patchified tokens. This makes masking more efficient by working on high-level representation instead of raw audio.
>
> **Span and causal masking along the channel axis.** The causal mask enforces a monotone ordering over channels: earlier channels must be sufficient to predict later ones. The span mask effectively acts as a low-pass filter on this sequence, suppressing local high-frequency fluctuations.
>
> Their combination yields smoother latents and consistently better diffusion-based generation.
>
> | Causal Mask | Span Mask | CLAP&uarr; | KL &darr; | FD &darr; |
> |---|---|---|---|---|
> | ✔ | | 0.28| 0.68| 176.04|
> |  | ✔ | 0.29|0.64|119.42|
> | ✔ | ✔ | **0.30** | **0.63** | **95.38** |
>
> **Rectified-flow decoder vs. ViT decoder.** We replace the ViT decoder with a rectified-flow denoising decoder, which iteratively predicts latents and reduces the train–test gap(as also noted in PeriodWave[5]). Ablations where we swap back to a ViT decoder show clear drops in KL and FD (with only minor CLAP gains), supporting this choice.
>
> | Latent Decoder | CLAP&uarr; | KL &darr; | FD &darr; |
> |---|---|---|---|
> | ViT decoder | **0.33** | 0.71| 126.27|
> | Denoise decoder | 0.29 | **0.69** | **114.50** |
>
> This indicates that:
> - the masking task alone already improves the latent space.
> - the flow decoder further refines latents to better match the VAE distribution used by the final decoder, yielding higher audio quality.

---

> > ### Author Response · Authors · 2025-12-02
> > **Response to Reviewer DYP6 (2)**
> >
> > ## 3 Typos and notation issues
> > We thank the reviewer for pointing these out and will correct them.
> >
> > ## 4 Clarifications on experimental results
> > > The presentation of the experiment section is confusing. Between row 4 and 5, is the gain due to LAFA or due to FT-AC? If MelVAE is better, why is LAFA needed? CLAP score seems to degrade in Table 2. In several cases of Table 3, seems like the gain from LAFA is almost negligible.
> >
> > ### 4.1 issues in Table1
> >
> > **Reimplementation gap to official SAO.**
> >
> > There is a misunderstanding in our original explanation of Table 1, and we appreciate the opportunity to clarify it. Our reimplementation of SAO underperforms the official SAO (in Table 3). However, we show the results of official SAO checkpoint in Table 1 and Table 2 for fair comparison.
> >
> > Our reimplementation of SAO underperforms the official SAO checkpoints mainly because:
> > - SAO does not release the exact captions used for training; we follow their guide to construct the captions, which presented in the Appendix B, but maybe a concern for the mismatched performance.
> > - Due to compute constraints, our DiT is trained on 30 s segments, whereas the official SAO model uses 47 s segments.
> >
> > Under these matched resource conditions, SAO‑VAE + LAFA beats our SAO baseline, which is the comparison we intend to highlight in Table 3.
> >
> > **Fine‑tuning on AudioCaps (FT‑AC).**
> >
> > AudioLDM2 and TANGO2 both include AudioCaps(AC) in their training sets. To compare fairly, we therefore also fine‑tune our DiT with LAFA on AC (DiT‑SAO‑FT‑AC). This substantially boosts CLAP and KL and outperforms AudioLDM2, indicating that the LAFA latent space scales well with better captions. We will clarify this in the table caption.
> >
> > We will also revise the text around Table 1–2 to state more carefully that LAFA consistently improves FD and KL over our SAO baseline under the same data and compute budget.
> >
> > ### 4.2 CLAP score in Table 2
> >
> > The lower CLAP score in Table 2 is largely attributable to differences in caption quality of our reimplemented baseline we discussed above.
> >
> > ### 4.3 Gains in Table 3
> >
> > The CLAP gains in Table 3 are indeed modest, which is expected given CLAP’s dependence on caption quality. This is consistent with Table 1, where CLAP improves notably after FT-AC with higher-quality AudioCaps captions. We will emphasize this metric distinction more clearly.
> >
> > ## 5 Why masking?
> > > I am trying to understand why the masking technique, in the space of design choices to address the problem, is the correct one. Or at least, why did authors decide to adopt this approach? I do see that masking is bringing the VAE output closer to the MEL behavior, so I
> >
> > Our goal is to increase the low-frequency components in the latent space and make it smoother. Inspired by MAE[5], the mask strategy enforces the decoder to do global interpolations to fulfill the reconstruction target, and the inter-patch topology will be increased by the attention mechanism. So, we intuitively take mask modeling to improve locality in latent space. And inspired by BERT[6], we found span masks are more efficient in representation learning. Moreover, we found it acts just like a convolution window in the masked sequence, that we prove it is a low-pass filter in latent space.
> >
> > ## References
> >
> > [1] Sang-gil Lee, Wei Ping, Boris Ginsburg, Bryan Catanzaro, and Sungroh Yoon. Bigvgan: A universal neural vocoder with large-scale training. arXiv preprint arXiv:2206.04658, 2022.
> >
> > [2] Yuepeng Jiang, Huakang Chen, Ziqian Ning, Jixun Yao, Zerui Han, Di Wu, Meng Meng, Jian Luan, Zhonghua Fu, and Lei Xie. Diffrhythm 2: Efficient and high fidelity song generation via block flow matching. arXiv preprint arXiv:2510.22950, 2025.
> >
> > [3] Roman Bachmann, Jesse Allardice, David Mizrahi, Enrico Fini, O˘guzhan Fatih Kar, Elmira Amirloo, Alaaeldin El-Nouby, Amir Zamir, and Afshin Dehghan. Flextok: Resampling images into 1d token sequences of flexible length. In Forty-second International Conference on Machine Learning, 2025
> >
> > [4] Dongchao Yang, Songxiang Liu, Haohan Guo, Jiankun Zhao, Yuanyuan Wang, Helin Wang, Zeqian Ju, Xubo Liu, Xueyuan Chen, Xu Tan, et al. Almtokenizer: A low-bitrate and semantic-rich audio codec tokenizer for audio language modeling. arXiv preprint arXiv:2504.10344, 2025
> >
> > [5] Kaiming He, Xinlei Chen, Saining Xie, Yanghao Li, Piotr Doll´ar, and Ross Girshick. Masked autoencoders are scalable vision learners. In Proceedings of the IEEE/CVF conference on computer vision and pattern recognition, pp. 16000–16009, 2022.
> >
> > [6] Jacob Devlin, Ming-Wei Chang, Kenton Lee, and Kristina Toutanova. Bert: Pre-training of deep bidirectional transformers for language understanding. In Proceedings of the 2019 conference of the North American chapter of the association for computational linguistics: human language technologies, volume 1 (long and short papers), pp. 4171–4186, 2019.

---

### Meta-Review · Area_Chair_DsmK · 2026-01-07

**Summary:**

This paper proposes a design of an audio latent space that is regularized to resemble Mel-spectrograms. However, the majority of reviewers raised concerns about the lack of ablations for key designs in the method and the confusing presentation, such as when comparing against SOTA approaches.

**Reviewer Concerns:**

Xq35: W1 about poor writing is acknowledged but not improved yet. W2 is clarified. W3 is not answered satisfactorily, in that the presented results use mixed setups that could confuse readers. Q1 is clarified, but the clarification does not resolve the concern about unfair comparisons.

hLGW: W1 is promised, but the fix is unverified. W2 is clarified, but does not resolve the concern about unfair comparisons. W3-5 are provided with additional experimental results, but the results do not fully support the arguments, e.g., ViT decoder has better CLAP than denoise decoder.

awW2: W1 response does not resolve the concern about qualitative results. W2-3 are provided additional experimental results, but not fully convincing, e.g., the result comparing span masking vs uniform masking remains missing. Q1-3 are explained.

DYP6: mostly addressed.

**Reviewer Scores:**

No changes

---

### Decision · Program_Chairs · 2026-01-26

Reject